# A Reconstruction of May–June Mean Temperature since 1775 for Conchos River Basin, Chihuahua, Mexico, Using Tree-Ring Width

Aldo Rafael Martínez-Sifuentes [1], José Villanueva-Díaz [1], Ramón Trucíos-Caciano [1,*], Nuria Aide López-Hernández [1], Juan Estrada-Ávalos [1] and Víctor Manuel Rodríguez-Moreno [2]

[1] National Institute of Forestry, Agricultural and Livestock Research, National Center for Disciplinary Research on Water, Soil, Plant and Atmosphere Relationships, Gómez Palacio 35150, Mexico; martinez.aldo@inifap.gob.mx (A.R.M.-S.); villanueva.jose@inifap.gob.mx (J.V.-D.); lopez.nuria@inifap.gob.mx (N.A.L.-H.); estrada.juan@inifap.gob.mx (J.E.-Á.)

[2] National Institute of Forestry, Agricultural and Livestock Research, National, Experimental Field Pabellón, Pabellón de Arteaga 20678, Mexico; rodriguez.victor@inifap.gob.mx

\* Correspondence: trucios.ramon@inifap.gob.mx

**Abstract:** Currently there are several precipitation reconstructions for northern Mexico; however, there is a lack of temperature reconstructions to understand past climate change, the impact on ecosystems and societies, etc. The central region of Chihuahua is located in a transition zone between the Sierra Madre Occidental and the Great Northern Plain, characterized by extreme temperatures and marked seasonal variability. The objectives of this study were (1) to generate a climatic association between variables from reanalysis models and the earlywood series for the center of Chihuahua, (2) to generate a reconstruction of mean temperature, (3) to determine extreme events, and (4) to identify the influence of ocean–atmosphere phenomena. Chronologies were downloaded from the International Tree-Ring Data Bank and climate information from the NLDAS-2 and ClimateNA reanalysis models. The response function was performed using climate models and regional dendrochronological series. A reconstruction of mean temperature was generated, and extreme periods were identified. The representativeness of the reconstruction was evaluated through spatial correlation, and low-frequency events were determined through multitaper spectral analysis and wavelet analysis. The influence of ocean–atmosphere phenomena on temperature reconstruction was analyzed using Pearson correlation, and the influence of ENSO was examined through wavelet coherence analysis. Highly significant correlations were found for maximum, minimum, and mean temperature, as well as for precipitation and relative humidity, before and after the growth year. However, the seasonal period with the highest correlation was found from May to June for mean temperature, which was used to generate the reconstruction from 1775 to 2022. The most extreme periods were 1775, 1801, 1805, 1860, 1892–1894, 1951, 1953–1954, and 2011–2012. Spectral analysis showed significant frequencies of 56.53 and 2.09 years, and wavelet analysis from 0 to 2 years from 1970 to 1980, from 8 to 11 years from 1890 to 1910, and from 30 to 70 years from 1860 to 2022. A significant association was found with the Multivariate ENSO Index phenomenon ($r = 0.40$; $p = 0.009$) and Pacific Decadal Oscillation ($r = -0.38$; $p = 0.000$). Regarding the ENSO phenomenon, an antiphase association of $r = -0.34$; $p = 0.000$ was found, with significant periods of 1 to 4 years from 1770 to 1800, 1845 to 1850, and 1860 to 1900, with periods of 6 to 10 years from 1875 to 1920, and from 6 to 8 years from 1990 to 2000. This study allowed a reconstruction of mean temperature through reanalysis data, as well as a historical characterization of temperature for central Chihuahua beyond the observed records.

**Keywords:** dendroclimatology; reanalysis models; ocean–atmosphere phenomenon

## 1. Introduction

The state of Chihuahua, located in northern Mexico, is known for its diverse geography and semiarid climate, characterized by extreme temperatures and marked seasonal variability; particularly, the central region of Chihuahua experiences climatic conditions that influence the temperature of the area, and is located in a transition zone between the Sierra Madre Occidental and the Northern Great Plain.

According to Mexico's National Meteorological Service, this area encounters hot summers and cold winters, with an average annual temperature ranging between 12 °C and 18 °C, depending on altitude and geographical location [1]. Climatic variability in central Chihuahua is also influenced by geographic factors such as altitude and topography, where higher elevations tend to experience colder temperatures, whereas valleys may experience thermal inversions that trap heat and elevate both daytime and nighttime temperatures [2].

The temperature variability in central Chihuahua holds significant socioeconomic and environmental implications, for instance, extreme temperature fluctuations can affect the health and wellbeing of the population, increasing the risk of heat and cold-related illnesses [3]. Additionally, climatic conditions can influence agricultural productivity, water resource availability, and regional biodiversity [2].

Climate station data in Mexico are limited (<75 years of records), and their reliability, quality, and geographical representation pose challenges for a comprehensive understanding of the inherent climatic variability across Mexico's various hydrological regions [4]. Currently, climatic databases provided by international institutions serve as optional sources of climate data available for a significant portion of North America. These databases rely on observed data and land surface models, offering diverse opportunities to acquire relevant environmental variables in spatial and temporal terms. These variables describe large-scale interactions of climatic dynamics and cycles that regulate local and regional conditions [5].

The advancement of modern technology, including automated equipment and satellite technology, has led to increased availability of climatic information over the past decade. This trend facilitates real-time quantification of climatic information [6]. An example of this is the NLDAS-2 model (North American Land Data Assimilation System v002), which provides monthly and hourly climatic data dating back to January 1979, with a spatial resolution of 0.125° [7]. Additionally, the CRU TS dataset (Climatic Research Unit gridded Time Series) consists of a high-resolution monthly grid of terrestrial observations (excluding Antarctica) dating back to 1901, comprising ten observed and derived variables [8]. Understanding historical patterns of hydroclimatic variability is essential for identifying trends and fluctuations over time. This knowledge informs the development of natural resource management plans and effective predictions, such as estimating water requirements for agriculture and predicting forest fires based on temperature changes.

In addition, this specific climatic reanalysis information allows the creation of regional climatic models with a great predictive capacity, robustness, and representativeness. This accumulated knowledge is fundamental for implementing adaptation strategies that address climatic variability and mitigate its impacts [9]. Annual tree rings are a fundamental tool for reconstructing past climate. The relationship between tree rings and climate derives from the fact that each year, trees form a new growth ring during the growing season (spring and summer in temperate regions). The formation of annual tree rings is influenced by climatic factors such as temperature, precipitation, and water availability. Favorable climatic conditions (moderate temperatures, adequate precipitation) during the growing season usually result in wider rings, as the tree can grow faster and accumulate more woody tissue. Conversely, unfavorable climatic conditions (extreme cold, drought, etc.) can lead to narrower rings due to slower or interrupted tree growth [10].

In the northern region of Mexico, dendroclimatic studies have been conducted to understand climatic dynamics over the last millennium and their interaction with atmospheric patterns [11,12], historical streamflow volume reconstruction [13], and fire regime reconstruction [14]. However, there is a lack of long-term data on historical temperature

variability in the central–northern region of Mexico, since they have only been developed for the northwest and northeast of the country [15,16].

The objectives of this study were (1) to generate an association between climatic variables for reanalysis (NLDAS-2 and CRU TS4.07 models) and earlywood chronology in central Chihuahua, (2) to generate a reconstruction of temperature, (3) to determine extreme events, and (4) to identify the influence of ocean–atmosphere phenomena with the temperature reconstruction.

## 2. Materials and Methods

### 2.1. Study Area

The Conchos River Basin (CRB) is located between 26°05′ and 29°55′ north latitude and 104°20′ and 107°55′ west longitude (Figure 1). It lies within the Hydrological Region 24 Río Bravo-Conchos, covering an area of 68,387 km$^2$ with elevations ranging from 772 to 3282 m.a.s.l. The CRB comprises five sub-basins: Conchos River-Ojinaga, Conchos River-La Colina Dam, Conchos River-El Granero Dam, Florido River, and San Pedro River [17]. The climate, according to the Köppen classification modified by García [18], is very arid in the Chihuahuan Desert region and semiarid to subhumid in the Sierra Madre Occidental. The temperature ranges from 8 °C to 18 °C in the upper part and from 16 °C to 22 °C in the lower part of the CRB, with an average annual precipitation of 419 mm [19].

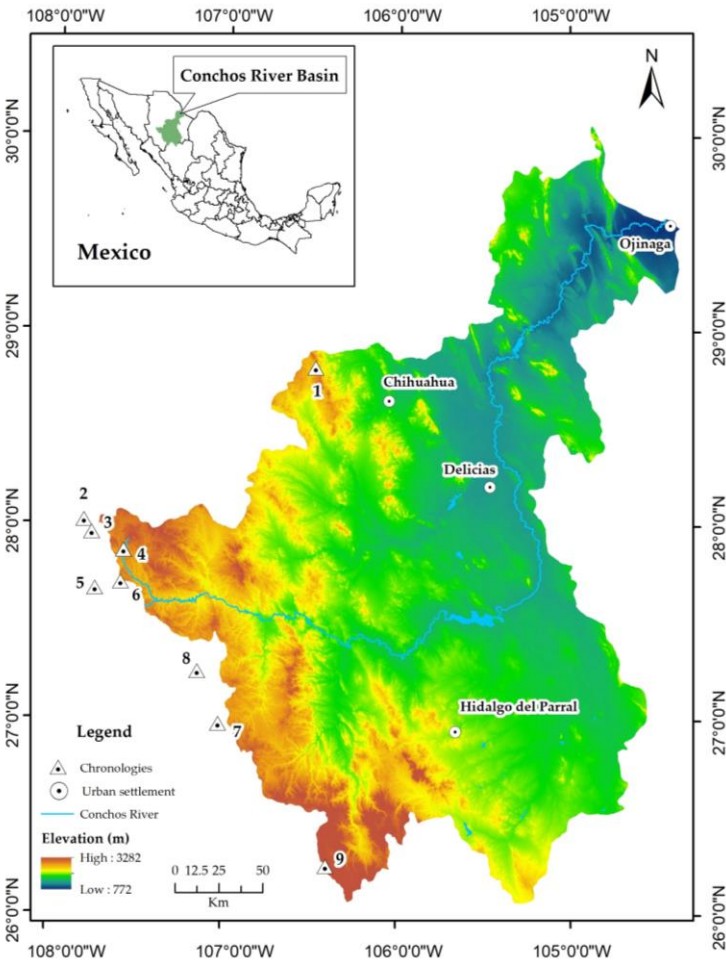

**Figure 1.** Geographical location of the nine chronologies on the periphery of the Conchos River Basin in Mexico.

### 2.2. Dendrochronological Information

Nine chronologies were downloaded from the International Tree-Ring Data Bank located within and on the periphery of the CRB (Figure 1, Table 1) [20]. Each chronology contains information on earlywood, latewood, and total ring width of the species *Pinus cembroides*, *Pinus arizonica*, *Pinus durangensis*, *Pinus lumholtzii*, and *Pseudotsuga menziesii*. The dendrochronological series for climate reconstruction were generated by the dendrochronological staff of the Dendrochronology Laboratory of the National Institute of Forestry, Agricultural and Livestock Research located in Mexico, and are available on the ITRBD portal (https://www.ncei.noaa.gov/access/paleo-search/?dataTypeId=18; accessed on 20 February 2024).

**Table 1.** Chronologies downloaded from the ITRBD for the Conchos River Basin.

| Number | Site | Code | Extension | Species [1] | Series Intercorrelation |
|--------|------|------|-----------|-------------|------------------------|
| 1 | Majalca | MAJ | 1750–2013 | PICE | 0.65 |
| 2 | Basagochi | CAC | 1809–2013 | PSME | 0.69 |
| 3 | Ranchito San Juanito | RAN | 1770–2013 | PICH | 0.54 |
| 4 | Baburiachi | BAB | 1889–2012 | PIAR | 0.60 |
| 5 | Barranca del Cobre | COB | 1745–2014 | PSME | 0.64 |
| 6 | Arareco | ARA | 1874–2014 | PIAR | 0.50 |
| 7 | El Tule Gpe. Y Calvo | ELT | 1830–2013 | PIDU | 0.54 |
| 8 | Guachochi | GUA | 1806–2017 | PILU | 0.62 |
| 9 | Los Pilares | LPI | 1725–2015 | PSME | 0.69 |

[1] PICE = Pinus cembroides, PSME = Pseudotsuga menziesii, PICH = Picea chihuahuana, PIAR = Pinus arizonica, PIDU = Pinus durangensis, PILU = Pinus lumholtzii.

The chronologies used in this study were developed using standard dendrochronological techniques [21], which include dating and measuring annual growth rings through the Velmex Inc. TA marking system with an accuracy of 0.001 mm. Dating control was performed using the COFECHA program considering a minimum intercorrelation of 0.328 ($p < 0.01$) between series [22].

Series standardization was generated with the ARSTAN program, through negative exponential curves and positive or negative slope lines, which allows the mitigation of the biological effect due to radial growth with age and natural or anthropogenic disturbances [23]. Standardization in the ARSTAN program helps to adjust the time series to eliminate or reduce the effect of nonclimatic factors that can influence the width of the rings, such as tree competition, tree age, and topographic position, among others. Therefore, researchers can perform more precise statistical analyses to identify climatic patterns, temporal trends, and relationships between the ring series of different trees and species, providing valuable information to better understand the response of trees to climate and how it has varied over time.

Dendrochronological statistics were determined for each series, such as interseries correlation (a measure of how well interannual ring width variability of one core matches with other cores), mean sensitivity (a relative change in ring width from one year to another), and autocorrelation (an influence that the growth of the previous year may exert on the growth of the current year) [24]. Expressed population signal (EPS; an indicator of correspondence between the variance of the chronology with the theoretical population) [25] and Rbar (a statistic used to examine the strength of the signal throughout the chronology) [26] were calculated using the dplR library with R software version 3.4.3 [27,28].

To maximize the climatic signal of the dendrochronological series and because correlations of 0.86 to 0.94 were found among the series of different species, a regional chronology was generated by combining all the individual series by band type, i.e., regional series of earlywood, latewood, and total rings.

### 2.3. Climatic Information

The climatic information was downloaded from the NLDAS-2 database, which is an offline data assimilation system that features uncoupled land surface models driven by observation-based atmospheric forcings with a grid spacing of 1/8 degree over North America since 1979 to the present [5]. Monthly accumulated precipitation (mm) and mean monthly temperature (°C) data were downloaded from 1979 to 2022 for the Conchos River Basin region. Relative humidity and maximum and minimum temperature data were downloaded from Climate Research Unit's (CRU TS4.07) ClimateNA reanalysis data with a spatial resolution of $1/2° × 1/2°$ from 1979 to the actual year [8].

The climatology of the study region is presented in Figure 2. Monthly mean precipitation is mainly concentrated in the summer months (July: 106.52 mm; August: 102.46 mm) due to convective rains originated by the North American Monsoon, contributing around 70% of the annual precipitation [29].

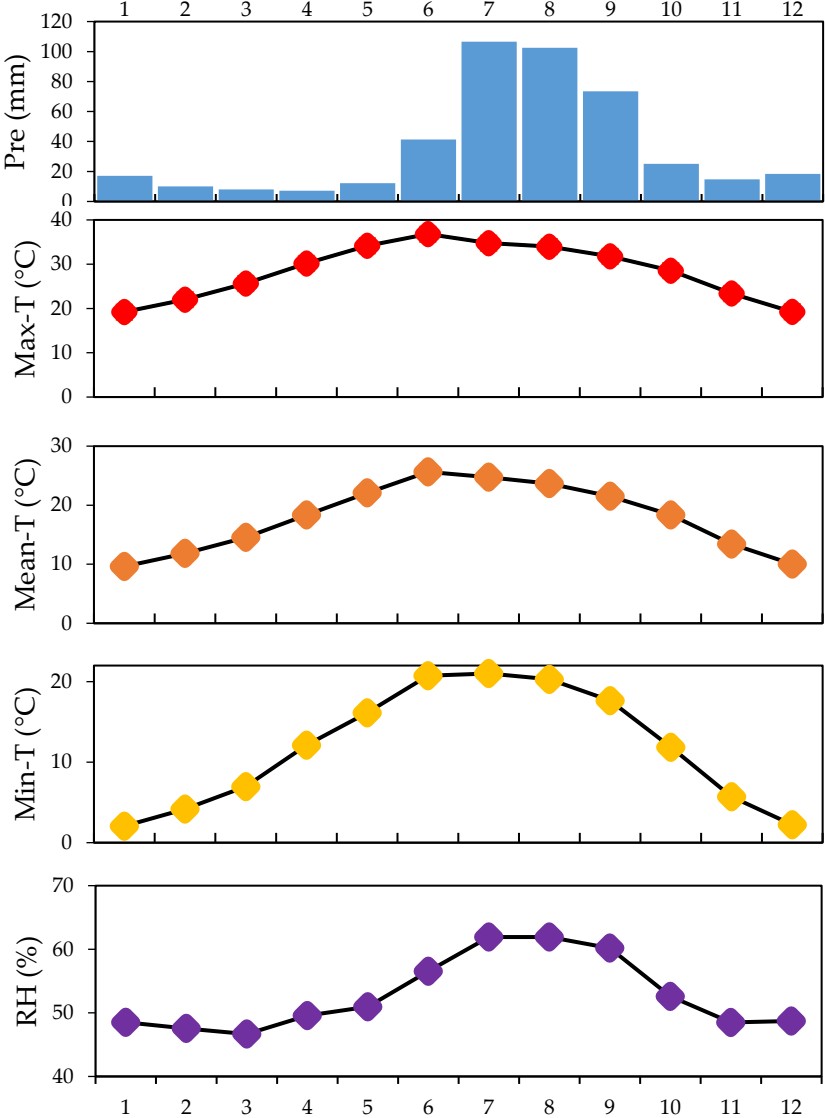

**Figure 2.** Multiannual information from 1979 to 2017 of monthly average precipitation, monthly maximum temperature, monthly average temperature, monthly minimum temperature, and monthly relative humidity percentage for the Conchos River Basin.

For the period 1979–2017, the highest average maximum temperature was found in June at 38.8 °C and the lowest in January at 19.2 °C; the lowest average minimum

temperature was recorded in January at 2.0 °C and the highest in July at 21.0 °C, with a historical average temperature of 17.8 °C. The highest percentage of relative humidity (RH) in the environment was recorded for August at 61.92% and the lowest for March at 46.66% (Figure 2).

### 2.4. Climate Association Analysis and Reconstruction

The association between climate variables (maximum temperature, minimum temperature, mean temperature, precipitation, and relative humidity) and regional series was generated using Pearson correlation ($p < 0.01$) at monthly and seasonal scales. For this analysis, seven months prior to the growth year and the following twelve months of the current year were considered [30] using the Dendroclim 2002 software [31].

For climate reconstruction, a linear regression model was developed with periods showing the highest and most significant correlation with climatic variables. The model was validated using the "verify" subroutine of the Dendrochronological Program Library of the University of Arizona, considering half of the records for calibration and the rest for model verification [22]. To validate the model, Pearson correlation parameters, Durbin–Watson coefficient, error reduction, and efficiency coefficient were considered using Minitab software version 17. To identify extreme periods, values outside the 95th percentile were considered for climate reconstruction.

To extend the time series, additional information after 2018 from the NLDAS-2 model was included. To evaluate the representativeness of the reconstruction, we employed spatial field correlation between the reconstructed climate series and ERA5 reanalysis data (a global climate reanalysis database that provides detailed information on various atmospheric and surface variables) for a self-defined region to investigate the spatial representativeness of the reconstruction over the observation period 1950–2022 with the KNMI (Koninklijk Nederlands Meteorologisch Instituut) Climate Explorer. This explorer allows users to access and visualize historical climate data and future climate projections for different regions of the world. Functions offered by the KNMI Climate Explorer include climate data visualization, climate trend analysis, climate projections, and data download, available on https://climexp.knmi.nl/start.cgi?id=someone@somewhere, accessed on 14 April 2024.

### 2.5. Periodicity and Ocean–Atmosphere Phenomenon Relationship in Climate Variability

A spectral analysis using the multitaper method was conducted to determine low-frequency event trends for the reconstructed series using the dplR library with R software version 3.4.3 [27]. This analysis enables the investigation of the influence of ocean–atmosphere phenomena on tree-ring growth [32]. Additionally, a wavelet spectral analysis was performed to identify significant periods in the reconstructed series using the biwavelet library with R software [27–33].

The influence of ocean–atmosphere phenomena on CRB temperature variability was determined using Pearson correlation analysis and the Multivariate ENSO Index (MEI; annual from 1979 to 2022) [34] and the Southern Oscillation Index (SOI; annual from 1951 to 2022) [35]. Additional analyses were conducted with the Atlantic Multidecadal Oscillation (AMO; annual from 1856 to 2022) [36] and the Pacific Decadal Oscillation (PDO; annual from 1854 to 2022) [37].

To determine the influence of the reconstructed variable and phase coherence in relation to ENSO, a wavelet coherence analysis of the common period between the reconstructed temperature and the ENSO 3.4 index was conducted [38] using the biwavelet package in R version 3.4.3 [33]. Wavelet coherence analysis is a technique used in signal processing and data analysis to study the relationship between two time series by combining wavelet theory with coherence analysis methods to examine how two signals are related on different time and frequency scales. That is, wavelet coherence allows the detection of common patterns between two data series that may not be evident with other analysis techniques, which is especially useful when working with data that have variability in time and frequency, such as nonstationary time series.

## 3. Results

### 3.1. Regional Dendrochronological Series and Climatic Association

The regional chronology begins from 1725; however, considering EPS > 0.85, it is limited from 1775 to 2017 (Figure 3). Only the standard earlywood (EW) regional chronology was considered for subsequent analyses because it showed the strongest response to climate and ocean–atmosphere phenomena compared to other regional chronologies (RW and LW chronologies).

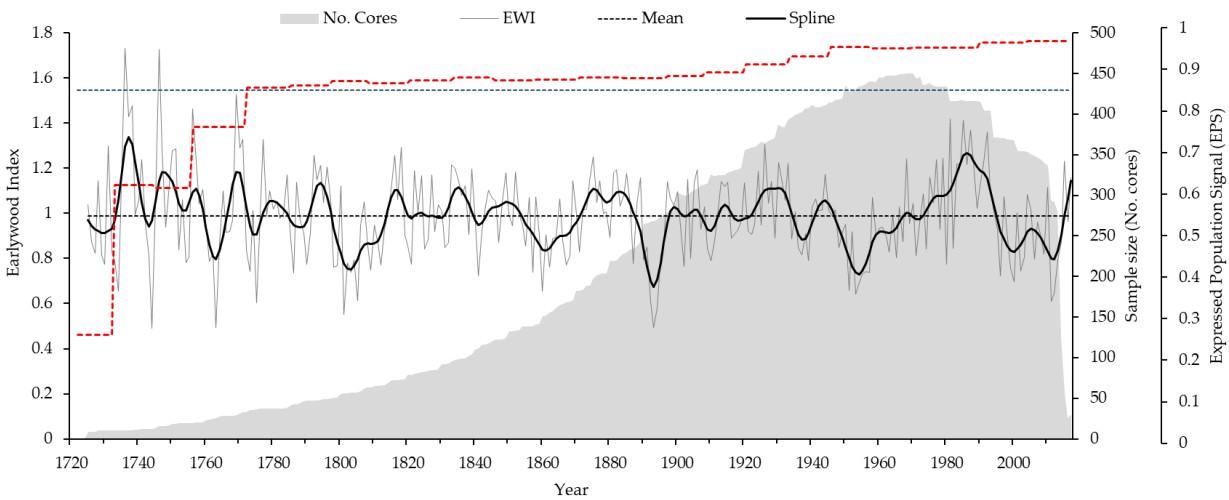

**Figure 3.** Regional earlywood chronology (gray line) with flexible decadal line to highlight low-frequency events (solid black line); gray shading represents the number of cores; EPS is represented by the red dashed line, and EPS > 0.85 is represented by the dashed blue line.

The Pearson correlation analysis between the regional earlywood chronology and the analyzed environmental variables showed highly significant periods ($p < 0.01$; Figure 4). For maximum temperature, high significance was found for the months of November of the previous growth year (−0.41), January of the current year (−0.60), and from March to June of the current growth year (−0.42, −0.54, −0.58, and −0.45).

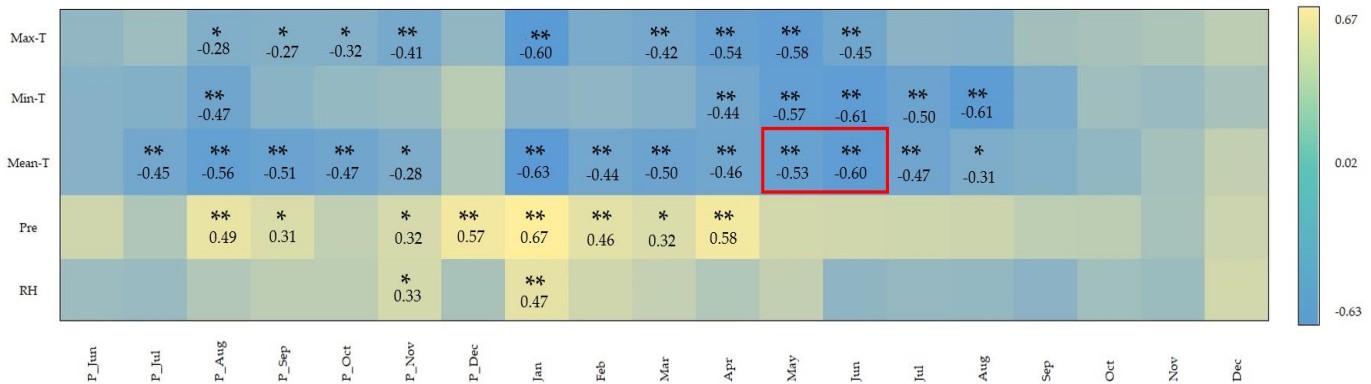

**Figure 4.** Pearson correlation analysis between regional earlywood chronology and climatic variables from 1979 to 2017 considering from June of the previous year to December of the current year. The seasonal period of mean temperature May–June ($r = -0.75$, $p < 0.01$) is represented within the red box. * Significant at 95%; ** highly significant at 99%.

Highly significant correlations with minimum temperature were found in the month of August of the previous year (−0.47) and from April to August of the current growth year (−0.44, −0.57, −0.61, −0.50, −0.61, and −0.38). Highly significant periods with mean temperature were from July to October of the previous growth year (−0.45, −0.56, −0.51,

and −0.47) and from January to July of the current growth year (−0.63, −0.44, −0.50, −0.46, −0.53, −0.60, and −0.47); however, it was found that the period from May to June of the current year showed a correlation of $r = -0.75$.

Highly significant correlations with precipitation were found in the months of August and December of the previous growth year (0.49 and 0.57) and from January to February (0.67 and 0.46) and in the month of April ($r = 0.58$). For relative humidity, significance was only found for the month of January ($r = 0.47$, $p < 0.01$) of the current growth year.

Additionally, significant associations ($p < 0.05$) were found with maximum temperature in the months of August to October of the previous growth year (−0.33, −0.32, and −0.32), and for minimum temperature in September of the current year (−0.38); for mean temperature in November of the previous year (−0.36) and August of the current year (−0.36); for precipitation in September and November of the previous year (0.38 and 0.33) and March of the current year (0.37); and for relative humidity in November of the previous growth year (0.32).

### 3.2. Temperature Mean Reconstruction

The mean temperature was reconstructed based on the seasonal period May–June ($r = 0.75$, $n = 39$, $p < 0.01$) (Figure 5).

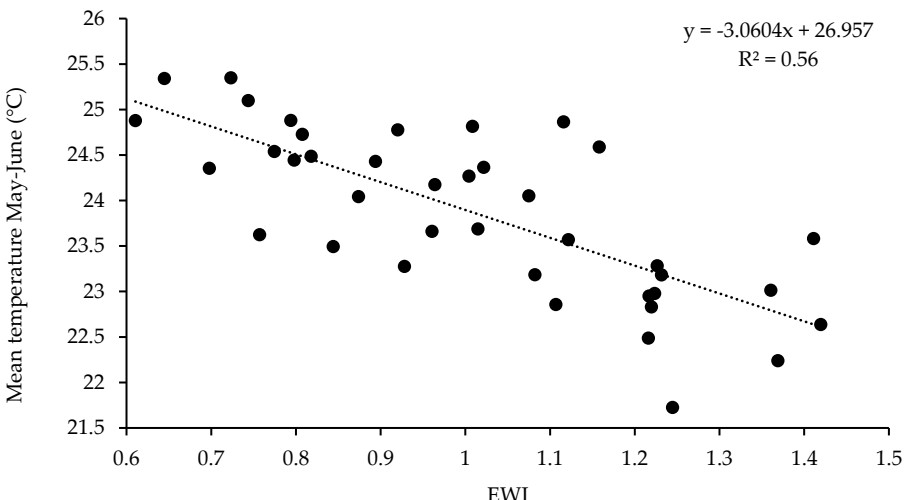

**Figure 5.** Association between mean temperature for the seasonal period May–June and earlywood index from 1979 to 2017.

The linear model for the period 1979–2017 is

$$Y = -3.0604 * X + 26.957 \tag{1}$$

where $Y$ represents the reconstructed mean temperature and $X$ represents the value of the earlywood index. The earlywood chronology explains 56% of the variance in mean temperature. The calibration and verification test was conducted with the subperiods of 1979–1997 and 1998–2017, respectively (Figure 6, Table 2).

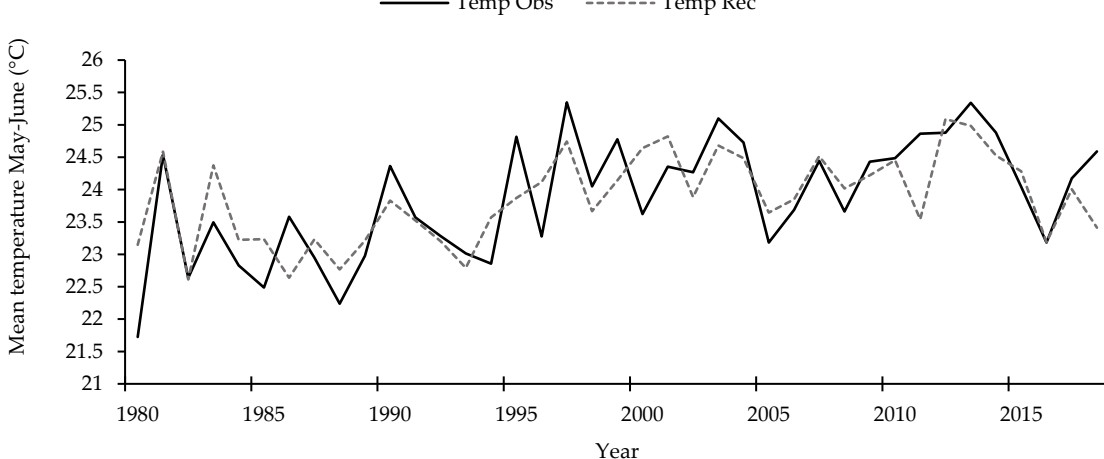

**Figure 6.** Association between observed and reconstructed mean temperature for the seasonal period May–June 1979 to 2017; 1979 to 1997 for calibration, and 1998 to 2017 for verification.

**Table 2.** Validation statistics for the linear model of mean temperature reconstruction.

| Statistics | Calibration (1979–1997) | Verification (1998–2017) |
| --- | --- | --- |
| Pearson correlation | 0.87 * | 0.54 * |
| Error reduction | 0.73 * | 0.75 * |
| "t" value | 3.69 * | 4.31 * |
| Signs test | 2 * | 2 * |
| First negative difference | 3 * | 3 * |
| Durbin–Watson coefficient | 2.32 | 1.67 |
| Efficiency coefficient | 0.48 * | 0.15 * |

* Significant ($p < 0.05$).

The reconstruction of mean temperature was extended from 1775 to 2017 using the earlywood chronology; however, the series was completed with the actual data from 2017–2022 from the NLDAS-2 model (Figure 7). Considering the 95th percentile, the events with the highest mean temperature were found in 1775 (25.1 °C), 1801 (25.2 °C), 1805 (25.0 °C), 1860 (24.9 °C), 1892–1894 (25.1, 25.4, and 25.1 °C), 1951 (24.9 °C), 1953–1954 (24.9 and 24.8 °C), and 2011–2012 (25.0, 24.9 °C). The flexible decadal line was adjusted to highlight low-frequency events.

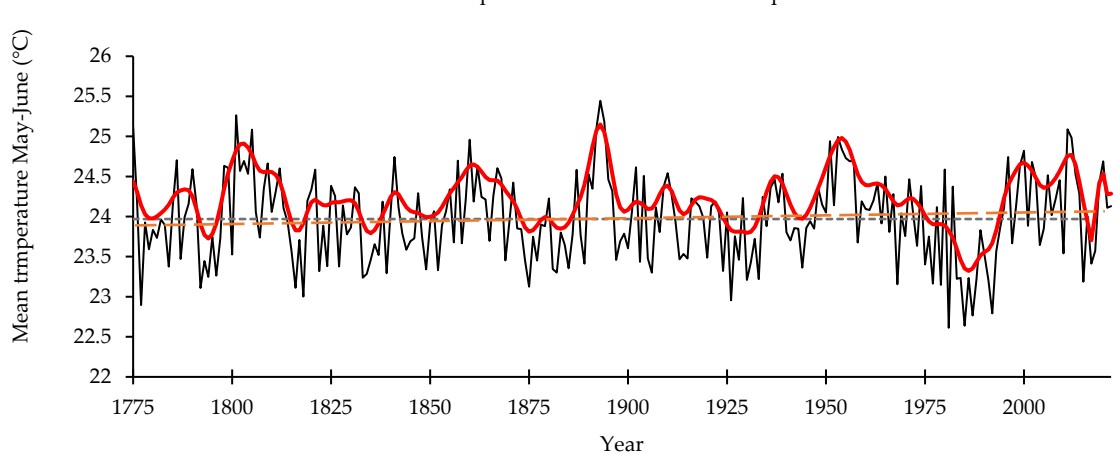

**Figure 7.** Annual mean temperature reconstruction for the period from 1775 to 2022 (black line) for the Conchos River Basin. The flexible decadal line highlights low-frequency events (red line), the dotted horizontal line is the reconstructed mean, and the orange dotted line is the upward trend.

The correlation between the regional earlywood chronology and temperature through spatial exploration (Figure 8) allowed the delineation of areas where temperature conditions had a significant influence ($p < 0.05$) on conifer growth in northern Mexico and western United States of America. The spatial representation of this spatial association showed a region with significant influence ($r > 0.6$) in northern Chihuahua and Coahuila, Mexico, as well as in New Mexico and Texas, United States of America.

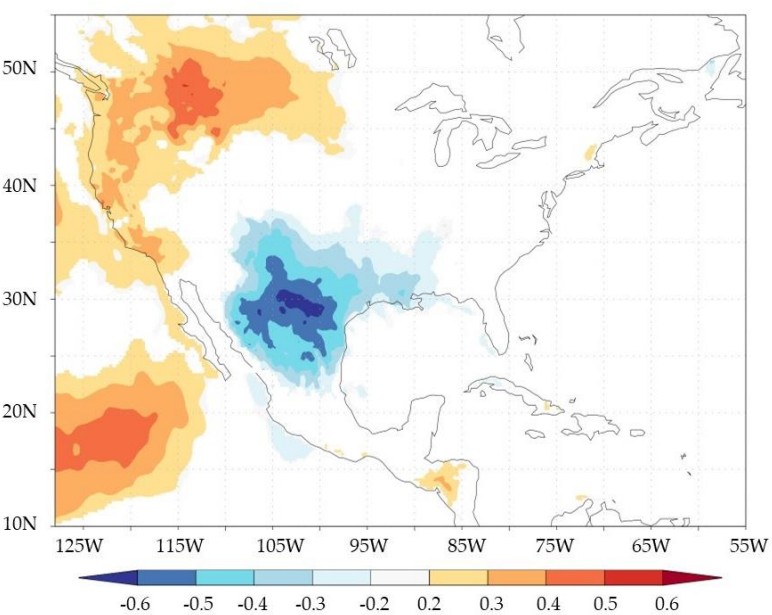

**Figure 8.** Spatial correlation between the regional earlywood chronology and seasonal mean temperature for May–June obtained from ERA5 reanalysis data with a resolution of $0.25° \times 0.25°$.

### 3.3. Spectral Analysis and Ocean–Atmosphere Phenomena Relationship

The spectral analysis using the multitaper method revealed significant cycles ($p < 0.01$) of 56.53 and 2.09 years, which are presented on interannual and multiannual scales (Figure 9A). These periods are corroborated by wavelet analysis, showing significant frequencies ($p < 0.05$) from 0 to 2 years from 1970 to 1980, from 8 to 11 years from 1890 to 1910, and from 30 to 70 years from 1860 to 2022 (Figure 9B).

The ocean–atmosphere phenomenon with the greatest influence on the mean temperature for the May–June seasonal period in the Conchos River Basin was in antiphase with the MEI ($r = -0.40$, $p = 0.009$, $n = 44$) and the PDO ($-0.38$, $p < 0.000$, $n = 169$). No significant associations were found for the SOI and AMO phenomena ($0.27$, $p < 0.022$, $n = 72$) and ($r = 0.11$, $p = 0.162$ $n = 167$), respectively.

The ENSO 3.4 phenomenon showed a significant association in antiphase ($r = -0.34$, $p < 0.000$, $n = 228$; Figure 10A), which is supported by Figure 10B, where various significant periods ($p < 0.05$) in antiphase are shown (arrows to the left); among them are periods of 1 to 4 years from 1770 to 1800, 1845 to 1850, and 1860 to 1900, with periods of 6 to 10 years from 1875 to 1920 and 6 to 8 years from 1990 to 2000.

**(A)**

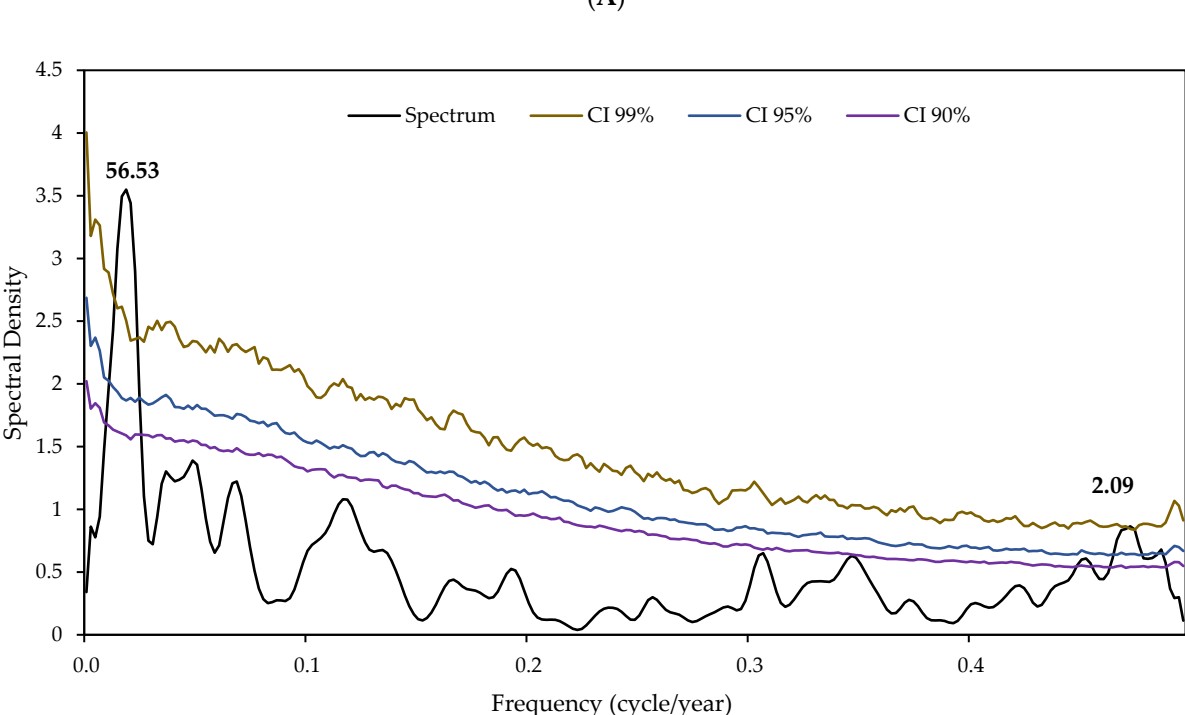

**(B)**

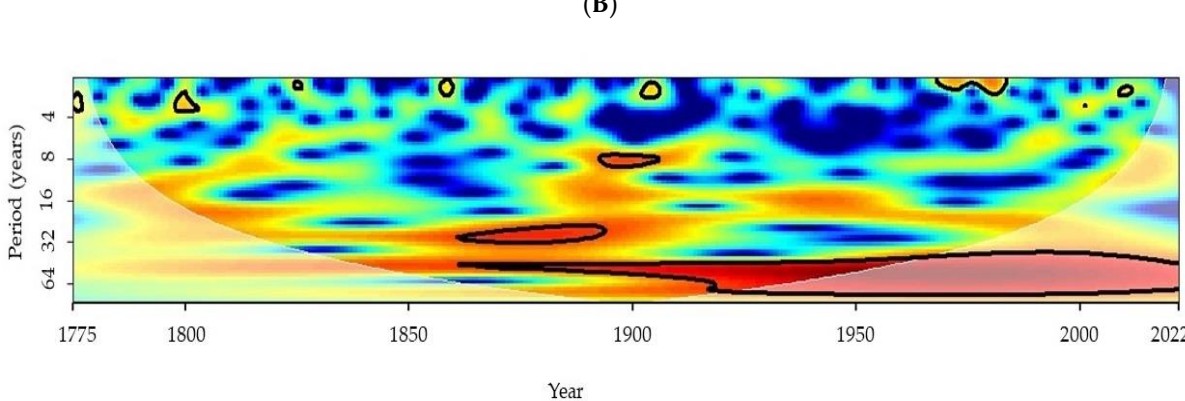

**Figure 9.** Multitaper spectral analysis of the reconstructed mean temperature of the Conchos River Basin at 90% (purple line), 95% (blue line), and 99% (brown line) confidence intervals (**A**). Wavelet analysis of the reconstructed mean temperature series; the thick black contours indicate the 95% significance level using the red noise model, and the cone of influence is in reddish color (**B**).

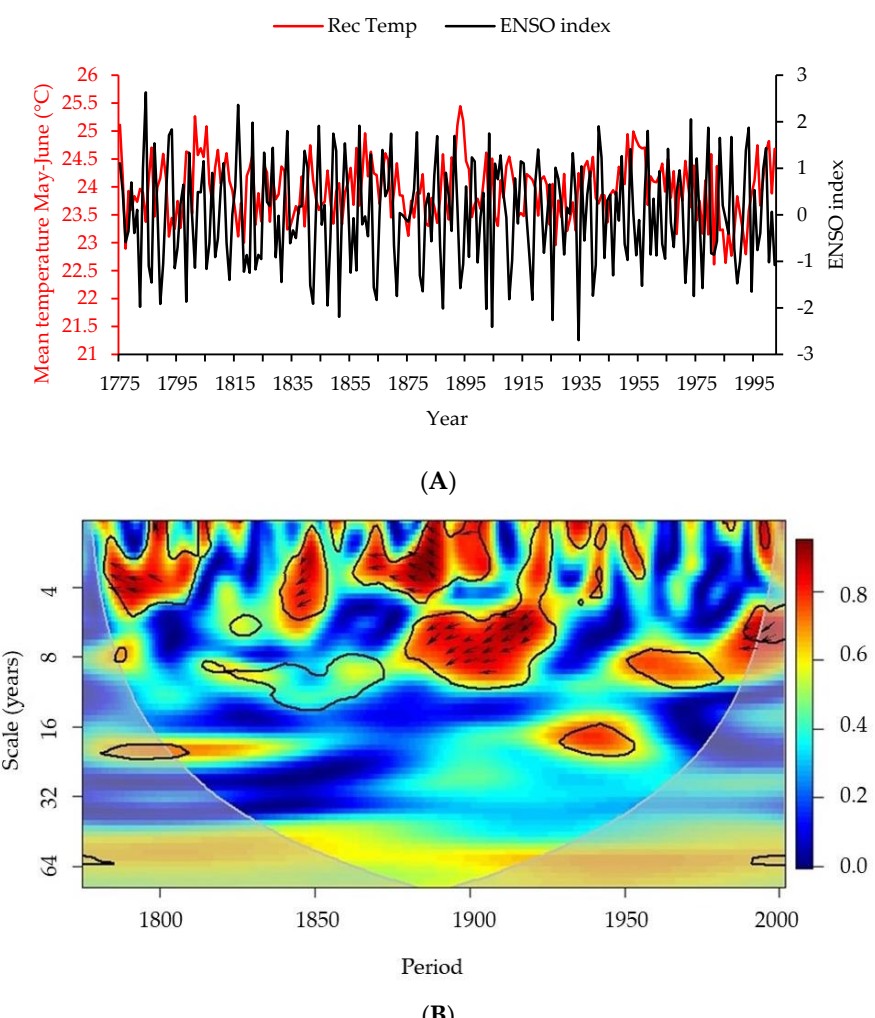

**Figure 10.** Association between mean temperature reconstruction and annual ENSO 3.4 from 1775 to 2002. Lines and labels in red correspond to temperature (**A**). Wavelet coherence analysis, red outlined areas correspond to significant periods, arrows on the left refer to antiphase; the thick black contours indicate the 95% significance level using the red noise model, and the cone of influence is in reddish color (**B**).

## 4. Discussion

### 4.1. Association of Environmental Variables with Earlywood Chronology and Temperature Reconstruction

The associations found between the regional earlywood chronology and precipitation in the Rio Conchos Basin region show significant relationships for December of the previous year and from January to April of the current year, periods of the winter–spring season that have been previously reported by other authors for northern Mexico [12–39].

The significant response to winter precipitation with earlywood growth is influenced by the warm phase of ENSO (El Niño) [35], a behavior reflected in conifer populations in the Sierra Madre Occidental [35–40]. Winter precipitation in northern Mexico accounts for 25% of the total annual precipitation. The winter season exhibits high interannual variability, although the presence of north winds favors winter rainfall and reduces temperatures in the northwest region of Mexico, increasing the incursion of polar air masses [41].

Conversely, spring precipitation is of great importance for species growth, which, being of low intensity and stored in the soil profile, is easily used by trees [42]. Additionally, wetter winters contribute to a better photosynthetic process in the warm season and are reflected in greater tree growth [43].

We found a positive and significant relationship between earlywood growth and relative humidity. This indicates that the accumulation of moisture during winter and spring is vital for early tree growth. Winter rainfall and ambient humidity increase water availability for trees and in the needles for the start of the growing season [38]. Relative humidity reflects the water vapor content of the plant–soil–atmosphere continuum; therefore, the level of relative humidity is directly related to soil moisture. Air relative humidity influences plant transpiration and photosynthesis by directly controlling stomatal conductance [44].

The relationship between monthly and seasonal temperature and ring width indices depends on the geographic location and limiting variables affecting tree growth [45], for example, the influence of maximum temperature on radial growth showed an inverse effect in most of the analyzed months, meaning that higher maximum temperatures resulted in lower growth, primarily from August to November of the previous year and during the spring and early summer of the current year (March to June). Significant relationships coinciding with those of the present study have been found for northern Mexico. For the Sierra Madre Oriental in the state of Coahuila, significant periods were found in July, August, and October of the previous year and from February to June of the current year [46]. For the Sierra Madre Occidental in the state of Durango, the same pattern of negative relationship with minimum temperatures and earlywood has been found; however, significant relationships were found only in March and April of the current year [47].

High maximum temperatures in summer increase evapotranspiration, causing water stress and reducing annual radial growth. This physiological behavior is common for tree species in the semiarid zones of northern Mexico and the southwestern United States, where precipitation is the most limiting factor for species growth [24]. Air temperatures above 25.0 °C drastically reduce stomatal opening, consequently reducing assimilation of photosynthates, leading to reduced radial growth [48].

Conversely, a negative influence of minimum temperature on radial growth was found for all analyzed months, but was significant during the summer season (May to August). Other studies have found a positive and significant relationship in January for the state of Durango [47], which differs from what was found in the present study. However, since it is in January where the lowest association was found, this may be because the minimum temperature in the state of Durango for that month averages −8.9 °C, compared to the 2.0 °C that the Conchos River Basin region in the state of Chihuahua can reach. The average minimum temperature in August in the northern region is 20.2 °C, which allows for a water vapor deficit in the needles and induces a reduction in stomatal opening [48]. The average summer minimum temperature and the high precipitation caused by the North American Monsoon interact to generate greater radial increments and vice versa [46].

In this study, the association between mean temperature and earlywood chronology was significant and negative, suggesting that temperature increases accelerate the evaporation process, depleting soil moisture in a shorter period and affecting the radial growth of conifer species [49]. The strongest association was found with the seasonal mean temperature of summer (May–June). This situation affects tree growth, where rising temperatures increase evapotranspiration, and if moisture availability is limited, it leads to increased vapor pressure deficits, elevating autotrophic respiration [50]. Vapor pressure deficit leads to higher evapotranspiration generated by an environment occurring during the premonsoon stage, compromising radial growth of conifer species [51].

The negative association between ring width growth and temperatures is globally significant, given the temperature increases associated with climate warming, which has become more evident in recent decades [12].

Dendroclimatic studies for northern Mexico have shown that EWs respond significantly to the summer season for drought studies [13] and precipitation reconstruction [15], because high temperatures favor increased evapotranspiration and alter the photosynthesis process, especially if we take into account that the conifer species in the current study have a Holarctic origin, with adaptations to low temperatures [13].

Recently, various reconstructions of seasonal precipitation [52–54] and runoff [46,55] have been developed for northern Mexico. However, reconstructions of temperature for this region with high climatic variability and the incidence of ocean–atmospheric phenomena are scarce. Among the generated studies are those of Villanueva-Díaz et al. [16] and Martínez-Sifuentes et al. [15], in which maximum temperature for the northeast and mean temperature for the northwest of Mexico, respectively, were reconstructed. The warm periods identified in the present reconstruction occurred in 1775, 1801, 1805, 1860, 1892–1894, 1951, 1953–1954, and 2011–2012. These periods coincide with temperature reconstructions in northwest Mexico [15]. Among these extreme periods, the one from 1951 to 1954 stands out, a period that, combined with scarce rainfall, reduced agricultural production and favored migration from rural areas to cities and to the United States of America [56]. High temperatures are associated with the incidence of forest fires; in this regard, there is coincidence with fire scars for the years 1953 and 2012 in the state of Durango [57]. Conversely, a cold period of 12 consecutive years was detected from 1983 to 1994 with an average of 23.2 °C and 0.7 °C below the reconstructed historical average, according to Villanueva-Díaz et al. [58]; this period is attributed to the influence of ENSO in its warm phase.

### 4.2. Spectral Analysis and Ocean–Atmosphere Phenomena Relationship

The effects of oceanic-atmospheric phenomena and their teleconnections are crucial for dynamic processes, such as seasonal wind patterns and variability in rainfall and temperature [59]. In this regard, the frequencies identified by spectral analysis using the multitaper method and wavelet analysis, ranging from 1 to 10 years, are situated within the domain of the ENSO phenomenon [60]. Specifically, these frequencies correspond to the "La Niña" phase, during which the most severe drought events in northern Mexico have occurred within these frequencies [61]. The ENSO phenomenon, reflected in this study through the ENSO 3.4 and the MEI, has given rise to intense events known as "MegaNiños" [62], some of which were identified in this analysis, resulting in social, economic, and environmental issues in northern Mexico [63].

The reconstructed temperature showed a significant correlation with the PDO ($r = -0.38$, $p < 0.000$, $n = 169$), indicating a possible link between this atmospheric phenomenon and climatic patterns that could affect fluctuations in mean temperature in northern Mexico [64]. The periods of drought that occurred in the 1930s are related to the positive phase of the AMO (periods above the reconstructed historical mean between 1936 and 1939). This trend is attributed to the presence of high pressure in the region, causing a decrease in the intensity of the low-level jet stream in the Gulf of Mexico [65]. The inverse relationship between the PDO and the AMO explains much of the climatic variability in northern Mexico [66]. Therefore, when the PDO is in a positive phase and the AMO is in a negative phase, wet episodes are generated; however, when the opposite occurs (negative PDO and positive AMO), episodes of drought are likely to prevail in this region [64].

### 5. Conclusions

In the present study, the feasibility and representativeness of using climate information generated by reanalysis models, along with tree-ring data, to extend temperature information over time was demonstrated. This approach is valuable for understanding climate variability in regions where meteorological station records are scarce, such as the central part of the state of Chihuahua in northern Mexico. The reconstruction of seasonal mean temperature showed interannual and multiannual variability, with the presence of periods considered very warm, such as 1775, 1801, 1805, 1860, 1892–1894, 1951, 1953–1954, and 2011–2012, with the year 1893 being the warmest in mean temperature over the last 248 years analyzed. These extreme periods negatively affected the social and economic progress of human populations in northern Mexico. For instance, the period coincided with the Industrial Revolution in Mexico, marked by consistently high summer temperatures. However, in the early 1950s, the reconstructed temperature increased compared to the

historical average, a trend that has become more notable in recent years. Temperature reconstruction is crucial to determine whether current changes are the result of natural variability or are largely influenced by changes in land use due to human activity or other phenomena such as the expansion of the agricultural and urban frontier, resource exploitation, or population pressure. In this study, an upward trend in temperatures was observed, which could be related to climate warming in northern Mexico.

**Author Contributions:** Conceptualization, A.R.M.-S. and J.V.-D.; methodology, A.R.M.-S., J.V.-D. and R.T.-C.; software, A.R.M.-S. and N.A.L.-H.; validation, J.V.-D. and J.E.-Á.; formal analysis, A.R.M.-S. and V.M.R.-M.; investigation, N.A.L.-H. and R.T.-C.; data curation, A.R.M.-S.; writing—original draft preparation, A.R.M.-S. and N.A.L.-H.; writing—review and editing, R.T.-C. and V.M.R.-M. All authors have read and agreed to the published version of the manuscript.

**Funding:** This research was funded by project of the Sectoral Research Fund for Education No. 283134/CB 2016-1 "Red dendrocronológica mexicana: aplicaciones hidroclimáticas y ecológicas".

**Institutional Review Board Statement:** Not applicable.

**Informed Consent Statement:** Not applicable.

**Data Availability Statement:** The data presented in this study are available on request from the corresponding author. The data are not publicly available due to privacy.

**Conflicts of Interest:** The authors declare no conflicts of interest.

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
