# Peer review of "A Reconstruction of May–June Mean Temperature since 1775 for Conchos River Basin, Chihuahua, Mexico, Using Tree-Ring Width"

_atmosphere, doi:10.3390/atmos15070808_

Round 1

Reviewer 1 Report

Comments and Suggestions for Authors

Author Response

Reviewer 1: Well written and a lot of good information is contained within the manuscript, but there is no “take-home point” as presented, which undermines the contribution of the research. Suggest a stronger focus on some aspect of the research. For example, is the key contribution the verification of the reanalysis data and why is that important? Likewise, the reconstructed climate data could be interesting, but listing extreme temperature years without a temporal analysis is underwhelming. From a synoptic climatology scale, what triggers the May-June extreme events? Is it surprising that in a warming world, there is no warming trend for this study area? Why might that be? Another potentially interesting point is that extreme May-June temperature events are more likely to be single-year, and do not extend beyond three consecutive years. Is this common/uncommon for other regions surrounding the study area?

Authors: The main contribution of the study is the reconstruction of the mean temperature through growth rings and reanalysis data, which is reflected in the objectives and emphasized. A temporal analysis is not performed, because it seeks to highlight the extreme years and discuss those periods. Through the ondeleta analysis, an answer is given on the extreme years and their relationship with the ocean-atmosphere phenomena. If a trend in the reconstruction of the temperature is appreciated.

Reviewer 1: Lack of detail with the variables. For lines 244-248, much more information is required. For example, what month(s) was (were) used for PDO, MEI, SOI, and AMO or was it annual data? If the latter, what are the beginning and ending dates?

Authors: Information on the variables and phenomena PDO, MEI, SOI and AMO was included (lines 286-291).

Reviewer 1: Most of the graphics are excellent, but the figure captions need more description of what the reader is viewing.

Authors: The description of all figures was increased.

Reviewer 1: Line 34: MEI not defined.

Authors: The MEI was defined.

Reviewer 1: Line 35: add p values to the two r-values.

Authors: p values werew added.

Reviewer 1: is “-0.34” an r-value

Authors: The r was inserted to define correlation.

Reviewer 1: Line 37: “allowed verifying” does that mean the study verified the effectiveness? More direct writing is needed.

Authors: the wording was changed (lines 40-42).

Reviewer 1: Line 45: delete “unique”

Authors: Done.

Reviewer 1: Lines 43–60 and elsewhere throughout the ms: There are a lot of short paragraphs of three or fewer sentences that make the reading choppy. Paragraph 153-156 is only one sentence. Suggest where applicable to merge short paragraphs. For example, merging two of the first three paragraphs.

Authors: The wording of the paragraphs has been improved.

Reviewer 1: Line 82: define “it”

Authors: Was defined in lines 84-86

Reviewer 1: Line 90: define “historical volume”

Authors: Done on line 88.

Reviewer 1: Line 102: delete “the extreme coordinates of”

Authors: Deleted

Reviewer 1: Line 105: change “meters above sea level” to “m a.s.l. or “m MSL”

Authors: Changed

Reviewer 1: Line 145: Suggest making species abbreviations four capital letters (e.g., Pce=PICE)

Authors: Suggestion accepted and abbreviations changed.

Reviewer 1: Lines 153-156: Dendro folks would know what this means, but unlikely others. Suggest elaborating. For example, what is the “biological effect due to radial growth.”

Authors: Increased the description of the process in lines 183-189.

Reviewer 1: Line 157: Interseries correlation is a measure of how well interannual ring width variability of one core matches with other cores.

Authors: Added as suggested by the reviewer (lines 191-192).

Reviewer 1: Line 182 and elsewhere: Use and “en” dash for ranges (e.g., Jan–Mar, 1999–2020) instead of a hyphen.

Authors: Done.

Reviewer 1: 183: For temperature, change to “38.8” and “19.2”. Change elsewhere applicable.

Authors: Done throughout the manuscript.

Reviewer 1: 217: Need more information in figure caption and “Climatic Information” should be lower case.

Authors: The description of the figure 2 was increased.

Reviewer 1: 249-252: Only a few readers will understand “wavelet coherence analysis”. Please add more information to simplify.

Authors: Increased the ondelete analysis information (lines 295-301).

Reviewer 1: 293: Figure 4 resolution is poor. And seasonal combinations are not shown. Adding r-values here would be helpful.

Authors: The resolution was improved and the values of r were added.

Reviewer 1: 333: Adding a trend line would be helpful for this figure.

Authors: Trend line added

Reviewer 1: 362: Figure A. Typo for “Frequency.” Need to explain/interpret what the reader is viewing in the figure caption.

Authors: The error was corrected and more description was included in the figure information.

Reviewer 1: 379: Why ENSO 3.0 instead of 3.4?

Authors: Thanks for the observation, the analysis was performed with ENSO 3.4, it was a typing error.

Reviewer 1: 397: Change “On the other hand” to “Conversely.” Same for line 427.

Authors: Done.

Reviewer 1: 409: One-sentence paragraph. Merge with another paragraph.

Authors: Done (lines 463-468).

Reviewer 1: 440: Change “but” to “and”

Authors: Done.

Reviewer 1: 451–463: Discussion of the absence of a warming trend would be an interesting contribution. Also, there is a focus on warm periods, but looking at Fig. 7, is the period from approximately 1975-1995, the coolest on record? Suggest commenting on that.

Authors: There is a trend and cold period information was added (lines 522-525).

Reviewer 2 Report

Comments and Suggestions for Authors

Review of the manuscript “A reconstruction of May-June Mean Temperature Since 1775 for Conchos River Basin, Chihuahua, Mexico Using Tree-Ring Width” by Martínez-Sifuentes et al The manuscript used available tree-ring chronologies and climatic parameters of reanalysis to develop their response functions to reconstruct May-June mean temperature in Conchos River Basin since 1775. It also applied spectral analysis and wavelet analysis into the reconstruction to determine its possible periods. This study should be of interest to the readers of Atmosphere. The current manuscript needs essential improvement, so I recommend a major revision. Major comments: 1. Tree-ring chronologies used in this study were downloaded from the International Tree Ring Data Bank. But I did not see their original sources and contributors as well as their original uses. Please tell the readers such information and let them have own assessments on the quality of tree-ring chronologies. 2. L336-354: I did not follow the authors’ intention to calculate spatial correlation between the regional earlywood chronology and seasonal mean temperature for May-June obtained from ERA5 reanalysis data. Please clearly elaborate why? how? in materials and methods (it would be better using “Data and methods”). At least tell readers it is for reconstruction or for association of regional earlywood chronology with large climatic patterns. BTW, say something about ERA5 reanalysis data. 3. L256-261: Using a table or Figure (such as Figure 8) to show the readers what you want to say. 4. L177: to the present, change to actual year. 5. L235: what is KNMI explorer? 6. L363: Figure 8 should be Figure 9. 7. L379: Figure 9->Figure 10

Comments on the Quality of English Language

no comments

Author Response

Reviewer 2: Tree-ring chronologies used in this study were downloaded from the International Tree Ring Data Bank. But I did not see their original sources and contributors as well as their original uses. Please tell the readers such information and let them have own assessments on the quality of tree-ring chronologies.

Authors: Added information on the chronologies used on lines 125-129.

Reviewer 2: L336-354: I did not follow the authors’ intention to calculate spatial correlation between the regional earlywood chronology and seasonal mean temperature for May-June obtained from ERA5 reanalysis data. Please clearly elaborate why? how? in materials and methods (it would be better using “Data and methods”). At least tell readers it is for reconstruction or for association of regional earlywood chronology with large climatic patterns. BTW, say something about ERA5 reanalysis data.

Authors: The reason for the spatial correlation and description of ERA5 was included (lines 269-278).

Reviewer 2: L256-261: Using a table or Figure (such as Figure 8) to show the readers what you want to say.

Authors: The description is located in the text, it is not possible to show in figure, since it does not represent something visual, only informative for the reader.

Reviewer 2: L177: to the present, change to actual year

Authors: Done.

Reviewer 2: L235: what is KNMI explorer?

Authors: Added on lines 274-278.

Reviewer 2: L363: Figure 8 should be Figure 9. 7. L379: Figure 9->Figure 10

Authors: Thanks for the observation, the figure numbers have been modified.

Reviewer 3 Report

Comments and Suggestions for Authors

Dear authors,

I think you have prepared a well-structured and properly written paper, about an interesting and not sufficiently studied region. Nevertheless, some dendroecological information is missing. Below you may find some general comments, followed by more specific suggestions.

General Comments

1.      Both in the abstract and elsewhere you use the term “extraordinary events”. It must be defined what this term means. Is it the same with extreme years?

2.      Figure 1 and Table 1 should be linked for the reader to know the location of each chronology within the study area

3.      It is not clear how you developed your regional chronology. Did you combine chronologies from different species and if yes what was the interspecies correlation? Given the fact that you have several different species and genus, this is crucial to be clarified and sufficiently justified before any further analysis

4.      Also it should be justified why you chose to use the earlywood chronology. Some information concerning the growing season should be provided. The justification in lines 257-260 is not sufficient from a dendro-ecological point of view.

Specific comments:

Line 61: consider using records instead of record.

Line 84: Better to replace “Tree growth rings” with “tree-ring” or “annual tree-rings”. Dendrochronologists do say tree-ring growth but the growth in the middle is not usual.

Line 86: replace “ancient” with “past”. Especially because the time span you are covering in the study has nothing to do with ancient.

Line 87: of course, there are previous and also (or more) important references that you could mention here. Luckman in 2010 was not the first who describe or discover the potential and usefulness of tree-rings. See Douglas for instance.

Lines 88-92: The paragraph needs better organization to avoid repetition.

Line 96: see general comment for extraordinary events

Line 102: why extreme coordinates?

Line 141: outside or near/ on the periphery

Lines 165-167: more information about the development of these chronologies, comprising many different species, is needed

Fig 2: you should give somewhere inside the text the abbreviation (RH)

Line 503: human-induced climate change is not only the result of land use changes

Author Response

Reviewer 3: Both in the abstract and elsewhere you use the term “extraordinary events”. It must be defined what this term means. Is it the same with extreme years?

Authors: The text was changed from "extremely" to "extreme" throughout the manuscript to homogenize the text and not confuse the reader.

Reviewer 3: Figure 1 and Table 1 should be linked for the reader to know the location of each chronology within the study área

Authors: Figure 1 and Table 1 were linked through the chronology numbers.

Reviewer 3: It is not clear how you developed your regional chronology. Did you combine chronologies from different species and if yes what was the interspecies correlation? Given the fact that you have several different species and genus, this is crucial to be clarified and sufficiently justified before any further analysis.

Authors: Correlation information added (lines 198-201).

Reviewer 3: Also it should be justified why you chose to use the earlywood chronology. Some information concerning the growing season should be provided. The justification in lines 257-260 is not sufficient from a dendro-ecological point of view.

Authors: Added justification in the discussion of using EW (lines 505-509).

Reviewer 3:   Line 61: consider using records instead of record.

Authors: Done.

Reviewer 3: Line 84: Better to replace “Tree growth rings” with “tree-ring” or “annual tree-rings”. Dendrochronologists do say tree-ring growth but the growth in the middle is not usual.

Authors: Done.

Reviewer 3: Line 86: replace “ancient” with “past”. Especially because the time span you are covering in the study has nothing to do with ancient.

Authors: Done

Reviewer 3: Line 87: of course, there are previous and also (or more) important references that you could mention here. Luckman in 2010 was not the first who describe or discover the potential and usefulness of tree-rings. See Douglas for instance.

Authors: Changed reference to:

Douglas, A.E. Crossdating in dendrochronology. Journal of Forestry, 1941, 39, 825-831. Available online: https://www.ltrr.arizona.edu/~ellisqm/outgoing/dendroecology2014/readings/douglass1941.pdf

Reviewer 3: Lines 88-92: The paragraph needs better organization to avoid repetition.

Authors: Reorganized paragraph (lines 96-101)

Reviewer 3: Line 96: see general comment for extraordinary events

Authors: The text was changed from "extremely" to "extreme" throughout the manuscript to homogenize the text and not confuse the reader.

Reviewer 3: Line 102: why extreme coordinates?

Authors: Removed "extreme coordinates", also suggested by another reviewer.

Reviewer 3: Line 141: outside or near/ on the periphery

Authors:  Added "on the periphery" on Figure 1 description.

Reviewer 3: Lines 165-167: more information about the development of these chronologies, comprising many different species, is needed

Authors: Correlation information added (lines 198-201).

Reviewer 3: Fig 2: you should give somewhere inside the text the abbreviation (RH)

Authors: Added in the text the meaning of RH (line 218).

Reviewer 3: Line 503: human-induced climate change is not only the result of land use changes

Authors: Conclusion increased (lines 253-268)

Reviewer 4 Report

Comments and Suggestions for Authors

    In this paper, the author used chronologies from the International Tree Ring Data Bank and climate data from the NLDAS-2 and ClimateNA reanalysis models to reconstruct mean temperatures from 1775 to 2022, identifying extreme periods and analyzing the influence of ocean-atmosphere phenomena like ENSO on temperature variations through spatial correlation, multi-taper spectral, and wavelet analyses.

Significant correlations were identified between tree-ring data and climate variables, particularly for the May-June mean temperature, with notable extreme periods and cycles detected, such as the 56.53 and 2.09-year frequencies, and associations with the MEI and PDO phenomena, validating the use of reanalysis data for climate reconstruction in the central Chihuahua region beyond available observational records. This paper provides an in-depth understanding of long-term temperature variability in the Conchos River Basin in Chihuahua and explores potential links to ocean-atmosphere phenomena, which will be valuble in studies of the region's climate history and in the development of future climate models.

1.     The abstract section should be updated to make it more logical. For example, in the first sentence, it should be stated why the research was conducted.

2.     In the Introduction section, authors did not provide detailed description on the relationship between tree-ring width and climates.

3.     P1-2: It is better to modify the content in L94-98 because that is same to those in the Abstract section.

4.     P3 Line 105: Delete the space for “3 282”.

5.     P11 L358-359 & 361: “Figure 8a” and “Figure 8b” in the text did not correspond to those in Figure 8, where a, b is capitalised. Please check and revise the entire text accordingly.

6.     P12 L373: “Figure 9a” and “Figure 9b” in the text also did not correspond to those in Figure 9, where a, b is capitalised. Please check and revise the entire text accordingly.

7.     P15 L489-505: I suggest refining content in Conclusion section. Those could be distilled into several gist to express.

8.     For Figure 1, 4, 8 and 9, please enhance their resolution.

9.     In Figure 2, please supplement the explanation on the horizontal and vertical coordinates. In addition, the space between five charts is not uniform, especially in the first two charts, please revise them accordingly.

Comments on the Quality of English Language

    I think the quality of English language is good. Minor editing is needed.

Author Response

Reviewer 4: The abstract section should be updated to make it more logical. For example, in the first sentence, it should be stated why the research was conducted.

Authors: The abstract began with the need for this study (lines 14-16).

Reviewer 4: In the Introduction section, authors did not provide detailed description on the relationship between tree-ring width and climates.

Authors: Information on the relationship between tree-rings and climate was included (lines 88-95).

Reviewer 4: P1-2: It is better to modify the content in L94-98 because that is same to those in the Abstract section.

Authors: The "objectives" section was modified to avoid duplication in the abstract and introduction section.

Reviewer 4: P3 Line 105: Delete the space for “3 282”.

Authors: Done.

Reviewer 4: P11 L358-359 & 361: “Figure 8a” and “Figure 8b” in the text did not correspond to those in Figure 8, where a, b is capitalised. Please check and revise the entire text accordingly.

Authors: Corrected the error in the figure number and changed it to capital letter.

Reviewer 4: P12 L373: “Figure 9a” and “Figure 9b” in the text also did not correspond to those in Figure 9, where a, b is capitalised. Please check and revise the entire text accordingly.

Authors: Corrected the error in the figure number and changed it to capital letter.

Reviewer 4: P15 L489-505: I suggest refining content in Conclusion section. Those could be distilled into several gist to express.

Authors: Conclusion improved

Reviewer 4: For Figure 1, 4, 8 and 9, please enhance their resolution

Authors: Improved resolution in Figures 1, 4, 8 and 9

Reviewer 4: In Figure 2, please supplement the explanation on the horizontal and vertical coordinates. In addition, the space between five charts is not uniform, especially in the first two charts, please revise them accordingly.

Authors: Figure 1 was modified

Round 2

Reviewer 1 Report

Comments and Suggestions for Authors

No comments.

Reviewer 2 Report

Comments and Suggestions for Authors

No more comments and suggestions.

Author Response

Thank you for your comments and suggestions 

Reviewer 3 Report

Comments and Suggestions for Authors

Dear authors,

Thank you for the revised version of the manuscript and for answering all the questions raised in the previous revision stage. I have only some minor comments/suggestions that you may find below:

Lines 88-90: “The relationship between tree-rings and climate, because, each year, trees form a new growth ring during the growing season (spring and summer in temperate regions).” I think something is missing in this sentence.

Line 90: you may replace “these rings” with “annual tree-rings”

Lines 98-99: “and fire regime reconstruction [14].” There are several other studies before the one you mentioned. E.g. https://doi.org/10.1139/x04-173 or https://doi.org/10.21829/abm41.1997.791. I do not know why you picked only this one.

Figure 1: why do you use capital for chronologies?

Line 199: among the series of different species? Please clarify

Line 274: KNMI. It would help if you gave the full name of this abbreviation.

Line 508: “in this study”. Do you mean “in the current study”?

Author Response

Reviewer: Lines 88-90: “The relationship between tree-rings and climate, because, each year, trees form a new growth ring during the growing season (spring and summer in temperate regions).” I think something is missing in this sentence.

Authors: The wording was corrected

Reviewer: Line 90: you may replace “these rings” with “annual tree-rings”

Authors: Done.

Reviewer: Lines 98-99: “and fire regime reconstruction [14].” There are several other studies before the one you mentioned. E.g. https://doi.org/10.1139/x04-173 or https://doi.org/10.21829/abm41.1997.791. I do not know why you picked only this one.

Authors: The reviewer's suggestion was accepted and the reference was changed.

Reviewer: Figure 1: why do you use capital for chronologies?

Authors: Due to a reviewer's suggestion, chronologies are additionally capitalized in other publications.

Reviewer: Line 199: among the series of different species? Please clarify

Authors: The wording was corrected

Reviewer: Line 274: KNMI. It would help if you gave the full name of this abbreviation.

Authors: done (Line 274).

Reviewer: Line 508: “in this study”. Do you mean “in the current study”?

Authors: The wording was corrected